# Risk Factors for Fungal Co-Infections in Critically Ill COVID-19 Patients, with a Focus on Immunosuppressants

**DOI:** 10.3390/jof7070545

**Published:** 2021-07-09

**Authors:** Obinna T. Ezeokoli, Onele Gcilitshana, Carolina H. Pohl

**Affiliations:** Yeast Research Group, Department of Microbiology and Biochemistry, University of the Free State, Bloemfontein 9300, South Africa or ezeokoliobinna@gmail.com (O.T.E.); GcilitshanaOMN@ufs.ac.za (O.G.)

**Keywords:** COVID-19, fungal co-infections, corticosteroid treatment, COVID-19-associated candidiasis, COVID-19-associated pulmonary aspergillosis, mucormycosis

## Abstract

Severe cases of coronavirus disease 2019 (COVID-19) managed in the intensive care unit are prone to complications, including secondary infections with opportunistic fungal pathogens. Systemic fungal co-infections in hospitalized COVID-19 patients may exacerbate COVID-19 disease severity, hamper treatment effectiveness and increase mortality. Here, we reiterate the role of fungal co-infections in exacerbating COVID-19 disease severity as well as highlight emerging trends related to fungal disease burden in COVID-19 patients. Furthermore, we provide perspectives on the risk factors for fungal co-infections in hospitalized COVID-19 patients and highlight the potential role of prolonged immunomodulatory treatments in driving fungal co-infections, including COVID-19-associated pulmonary aspergillosis (CAPA), COVID-19-associated candidiasis (CAC) and mucormycosis. We reiterate the need for early diagnosis of suspected COVID-19-associated systemic mycoses in the hospital setting.

## 1. Introduction

The current COVID-19 disease has infected over 180 million people worldwide, with associated deaths in upwards of 3.9 million [1]. The disease is characterized by a variety of symptoms, of which breathing difficulties are typical of severe cases [2]. Such severe cases require urgent intervention in hospitals, including oxygenation and mechanical ventilation. For such patients, a variety of other complications may arise, including hospital-acquired secondary infections with opportunistic pathogens, including molds and yeast infections. In the current severe acute respiratory syndrome-coronavirus-2 (SARS-CoV-2) pandemic, fungal co-infections among COVID-19 patients in intensive care units (ICUs) has been reported [3,4,5,6,7]. White et al. [7] reported an incidence of 14.1% and 12.6% for aspergillosis and yeast infections, respectively, amongst critically ill COVID-19 patients across multiple centers in Wales. Bartoletti et al. [6] reported a higher incidence of 27.7% for invasive pulmonary aspergillosis (IPA) (or COVID-19-associated pulmonary aspergillosis, CAPA) amongst COVID-19 patients (of a 108 patient cohort) requiring invasive mechanical ventilation between late February and April 2020 in Italy. Based on analysis of clinical data from several countries, Salmanton-Garcia et al. [5] found that the aggregate incidence of CAPA was between 1% and 39.1% amongst COVID-19 patients in ICUs. Apart from CAPA, the incidence of candidemia has also been reported among hospitalized COVID-19 patients, with up to 12% (106/889) reported for one health center [8] and between 1.54% and 7.54% for a hospital in Rio de Janeiro, Brazil [9]. Indeed, the observations of fungal co-infections, especially IPA, are not entirely unexpected in the current SARS-CoV-2 outbreak, given that similar observations were made during previous outbreaks of other coronaviruses such as the severe acute respiratory syndrome-coronavirus (SARS-CoV) and the middle east respiratory syndrome coronavirus (MERS-CoV) [10,11,12].

In order to elucidate the diversity and co-occurrence of fungal co-infections in COVID-19 patients, we summarized several randomly selected reports on fungal co-infections in COVID-19 patients (Figure 1). These reports span all but one of the inhabited continents, Africa, where to the best of our knowledge, little or no reports of fungal co-infections in critically ill COVID-19 patients currently exists. Based on our non-exhaustive summary of literature reports, at least 20 different fungal species have been reported in hospitalized COVID-19 patients. The majority of fungal co-infections are due to *Aspergillus fumigatus* (most common etiological agent of CAPA), followed by *Candida albicans* (common etiological agent of candidiasis or candidemia) (Figure 1a). Furthermore, we observed significant co-occurrences of *Candida albicans* and other fungal pathogens such as *Aspergillus fumigatus*, *A*. *flavus* and *A*. *penicillioides* amongst the reports (Figure 1b).

In this review, we reiterate the role of fungal co-infections in exacerbating COVID-19 disease severity as well as highlight emerging trends related to fungal disease burden and multidrug resistance in COVID-19 patients. In particular, we provide up to date perspective on the risk factors for systemic mycoses in hospitalized patients with a specific focus on the potential role of immunomodulatory and/or immunosuppressive drugs in driving the observed high prevalence of fungal co-infections in COVID-19 patients in the ICU.

## 2. Fungal Co-Infections in COVID-19 Disease: Disease Severity, the Emergence of Multidrug Resistance and Mucormycosis

### 2.1. Contribution to COVID-19 Disease Severity and Mortality

Prospective and retrospective data of COVID-19 patients admitted to intensive care units (ICU), especially for a prolonged duration, show that these patients are susceptible to invasive microbial co-infections during hospitalization and that these may lead to more severe outcomes [3,7,16,31]. A prospective cohort study of 135 adults, performed across multiple centers in Wales, showed a significantly higher (up to 25%) mortality rate in COVID-19 patients with fungal infections compared to patients without fungal infections [7]. In particular, a multicenter study of 108 COVID-19 patients admitted to the ICUs in Italy showed a significantly higher 30-day mortality rate for patients with probable COVID-19-associated pulmonary aspergillosis (CAPA) or putative invasive pulmonary aspergillosis, compared to patients without suspected aspergillosis [6]. Similarly, Meijer et al. [38] reported mortality between 40% and 50% in patients with CAPA across the first wave (March–April 2020) and second wave (mid-September to mid-December 2020) of the COVID-19 pandemic in Brazil.

With respect to COVID-19-associated candidiasis (CAC), although the incidence rates may be slightly lower than that of CAPA, the mortality rate of COVID-19 patients with candidemia do not appear to differ markedly. Reports from Italy indicate that up to 57.1% and 50% mortality was reported in COVID-19 patients with candidemia [57] and *Candida auris* candidemia [27], respectively. Elsewhere, the mortality rate of COVID-19 patients with candidemia exceeded those of counterparts without candidemia in Iran (100% vs. 22.7%) [31]. Altogether, these observations highlight and re-emphasize the propensity of fungal co-infections to exacerbate disease severity and, consequently, increase the mortality of critically ill patients admitted to the ICUs.

### 2.2. Emergence of Multidrug-Resistant Fungi

Another concerning development is the report of the incidence of multidrug-resistant *Aspergillus fumigatus* [56] and *Candida auris* [26,27,58], as well as pan-echinocandin resistant *C*. *glabrata* amongst COVID-19 patients [59]. *C. auris* is a multidrug-resistant fungal pathogen that causes life-threatening systemic infections and could have a 30-day mortality rate of up to 35% [60,61,62] and a report from India stated a case-fatality rate of 60% amongst COVID-19 patients with candidemia due to multidrug-resistant *C*. *auris* infection [26]. Managing candidemia outbreaks due to *C*. *auris* poses a great challenge due to its resistance to multiple drugs and persistence in the human body and environment [63,64]. Suarez-de-la-Rica et al. [65] observed that co-infections by antimicrobial-resistant pathogens might be consequential for COVID-19 prognosis. In their study, the hazard ratio for death within 90 days in critically ill COVID-19 patients was significantly increased by antimicrobial-resistant pathogens. Hence, given that antifungal resistance undermines treatment efforts and can escalate treatment costs, such reports of multidrug resistance to antifungals, including echinocandins, are worrying. Furthermore, these reports highlight the need to appreciate the global burden of fungal co-infections in the current COVID-19 pandemic and the importance of prompt diagnosis and treatment of fungal pathogens in hospitalized COVID-19 patients.

### 2.3. Incidence of Mucormycosis

Recent reports emanating from India indicate the incidence of mucormycosis mostly among COVID-19 survivors (although cases in currently hospitalized COVID-19 patients have also been observed) [66,67,68,69]. Mucormycosis is a rare infection caused by filamentous fungi (molds) of the order *Mucorales* and can be fatal if the fungus penetrates the central nervous system [70,71]. According to one study, *Rhizopus arrhizus* is currently the most common etiological agent of COVID-19-associated mucormycosis (CAM) in India, with *Rhizopus microsporus*, *Rhizopus homothallicus*, *Mucor irregularis*, *Saksenaea erythrospora* and *Apophysomyces variabilis* also implicated in some cases in India and elsewhere [72,73,74]. Mucormycosis infections are common in diabetic or immunocompromised patients, including persons receiving immunosuppressive therapy [71,75]. A recent systematic review of mucormycosis cases in India and worldwide reported that corticosteroid use was recorded in 76.3% of cases and that 30.7% of mucormycosis cases were fatal [76]. Thus far, it is unclear whether the incidence of CAM is worldwide, and research is needed in this regard. Altogether, the foregoing reports and several other studies suggest that the heightened incidence of mucormycosis in India are related to certain risk factors, including poorly managed diabetes and the prolonged usage of high dosage steroids in treating COVID-19 [69,77,78].

## 3. Overview of Risk Factors for Opportunistic Fungal Infections in Critically Ill COVID-19 Patients

Overall, risk factors driving the high incidence of aspergillosis and candidemia in COVID-19 patients are related to invasive procedures (e.g., intubation) predisposing lung tissues to fungal colonization and proliferation [79,80,81], history of chronic pulmonary disease [7], prolonged corticosteroid treatments [7,82], immunological disposition of patients and antimicrobial therapy [59,64]. In one study comparing co-infections in critically ill patients with and without COVID-19, it was observed that the need for invasive assisted respiration was the most decisive factor for co-infections with antifungal-resistant pathogens in patients with severe COVID-19 [79]. In the following subsections, we provide an overview of risk factors associated with CAPA and CAC—the two most commonly reported fungal co-infections in hospitalized COVID-19 patients.

### 3.1. Risk Factors for CAPA

Given the high incidence of CAPA and the distinct clinical features of CAPA compared to influenza-associated pulmonary aspergillosis, it has been necessary to establish appropriate case definitions for CAPA in order to facilitate uniformity of reporting across medical practices. To this end, a number of case definitions or guidelines have been proposed for characterizing possible, putative, probable and proven CAPA cases [7,83,84] (case definitions for CAPA is not within the purview of this review). CAPA is essentially defined as pulmonary or tracheobronchial infection with *Aspergillus* spp. in COVID-19 patients. In one proposed case definition by Koehler et al. [83], a proven case of CAPA may be established by direct microscopic and/or histopathological evidence of fungal features that are typical of *Aspergillus* spp. Such evidence includes an observation of invasive growth into tissues with concomitant tissue damage, recovery of *Aspergillus* spp. by culture, detection of *Aspergillus* by microscopy in histology studies or by polymerase chain reaction from materials obtained by sterile aspiration or biopsies from a pulmonary site indicating an infectious disease.

Several factors predispose hospitalized COVID-19 patients to CAPA. SARS-CoV-2 insults in the lungs elicit the release of danger-associated molecular patterns (DAMPs) in severe COVID-19 [80]. Essentially, DAMPs are host-derived molecules that regulate the activation of pathogen recognition receptors and modulate the host’s organ response to injury [85]. The release of DAMPs is accompanied by inflammation and extensive damage of lung epithelial tissues, which are enabling risk factors for invasive pulmonary aspergillosis [80]. Other pathophysiological factors identified for IPA and potentially CAPA, include an impaired local immune response and a dysfunctional defective mucociliary activity [12,50]. In severe COVID-19 disease, extensive inflammation and injury to the lungs lead to acute respiratory syndrome (ARDS). ARDS is characterized by difficulty in breathing; hence, assisted ventilation is required for such patients. However, mechanical ventilation and the duration of ventilation is a known risk factor for invasive aspergillosis and CAPA in the ICU [50,86,87].

In addition, pharmaceutical treatments for malignancy and the use of corticosteroids (discussed in a later section) and antibiotics may be risk factors for CAPA [7,88]. For example, in a multicenter study across Wales [7], a significant association was observed between COVID-19 patients with IPA and patients treated for or diagnosed with solid malignancy. Further, in a multilocation retrospective cohort conducted in France, Dellière et al. [88] reported that treatment with azithromycin for up to 3 days significantly correlated with the incidence of probable invasive pulmonary aspergillosis in COVID-19 patients. Such observation was attributed to the immunomodulatory properties of azithromycin that may weaken the host’s immune response and subsequent susceptibility to aspergillosis [88,89].

### 3.2. Risk Factors for CAC

COVID-19-associated candidiasis (CAC) refers to the detection of one or more *Candida* spp. in the bloodstream or body tissues of COVID-19 patients. As earlier indicated, *Candida albicans* and other non-*albicans Candida* species have been reported among hospitalized COVID-19 patients [9,29,57,81]. Some of the risk factors identified for CAC include prolonged hospital stays, mechanical ventilation, central venous catheters, surgical procedure, and the use of broad-spectrum antibiotics [9,29]. For example, Nucci et al. [9] observed that COVID-19 patients with candidemia were more likely to be under mechanical ventilation than non-COVID-19 patients. Similarly, Mastrangelo et al. [57] reported that COVID-19 patients with candidemia were more likely to be in the ICU and receiving immunosuppressive agents than patients in the ICU for reasons other than COVID-19. In another report, the development of pan-echinocandin resistant *C*. *glabarata* co-infections in a hospitalized 53-year-old COVID-19 patient potentially rendered antifungal treatment ineffective and probably aggravated the progression of the disease [59]. According to the authors, the case report demonstrated the need for active monitoring for antifungal resistance development in order to inform the dynamic use of effective antifungal agents during patient management in the intensive care unit.

## 4. Immunosuppressants as Risk Factors for Fungal Infections in Critically Ill COVID-19 Patients

Most of the current treatment options for managing patients with severe COVID-19 are immunomodulators [90]. The anti-inflammatory properties of these immunomodulators are important to counteract the heightened and unregulated release of pro-inflammatory cytokines (also known as ‘cytokine storm’) in the lungs during SARS-CoV-2 infection [91,92]. Thus, immunosuppressants such as dexamethasone, methylprednisolone, prednisone, hydrocortisone and tocilizumab constitute the most common treatment options for managing severe COVID-19 cases in the ICU [90]. From a COVID-19 treatment standpoint, immunostimulants are required during the early stages of the disease, whereas immunosuppressants may be more beneficial to dampen the cytokine storm in the later stages of the disease [93]. For example, dexamethasone treatment decreased the 28-day mortality in COVID-19 patients on invasive respiratory support or receiving oxygen alone but was not particularly beneficial for COVID-19 patients with less severe disease, suggesting that hyper inflammation mediates the advanced stage of the disease and therefore can be alleviated by immunosuppressants [93,94].

Unfortunately, the immunosuppressants hamper both the individual’s innate and adaptive immune responses through sophisticated quantitative and qualitative mechanisms of immune deregulation [88,95,96,97,98], thereby increasing patients’ susceptibility to invasive fungal diseases. In particular, steroidal immunosuppressants such as corticosteroids predominantly affect the protective immunity process qualitatively through functional impairment of several effector immune cells, such as monocytes, polymorphonuclear leukocytes, T lymphocytes and macrophages [96] and is a significant acquired immunological risk factor for pulmonary aspergillosis [99,100]. Thus, corticosteroids such as dexamethasone and methylprednisolone, used for managing critically ill COVID-19 patients, have contraindications, including fostering secondary microbial infections in patients [99,101].

In the present COVID-19 pandemic, questions are being asked regarding the possible relationship between immunosuppressant or corticosteroid use and the incidence of fungal infections in critically ill COVID-19 patients (Table 1). A prospective study conducted across multiple COVID-19 intensive care units in Wales showed that the use of high-dose systemic corticosteroids significantly increased the odds of COVID-19 patients developing aspergillosis [7]. Similarly, in Brazil, Riche et al. [82] observed a 10-fold increase in candidemia amongst a cohort of critically ill COVID-19 patients receiving high doses of corticosteroids such as prednisone, hydrocortisone, methylprednisolone and dexamethasone, while a retrospective study conducted in Chicago and involving 111 COVID-19 patients receiving tocilizumab (a monoclonal antibody that inhibits binding of IL-6 to the membrane and soluble receptors [102]) was significantly linked with the risk of developing fungal pneumonia and sinusitis [103]. However, in a retrospective study involving 4313 COVID-19 patients in New York, corticosteroid use was not associated with increased bacteremia or fungaemia compared to non-corticosteroid users when administered within the first seven days of admission [104]. In this study, the early administration of low dose corticosteroids is advocated. It must, however, be noted that in many of the other reports, mention is made of high doses of corticosteroids often administered for prolonged periods [30], which may explain the high incidence of systemic fungal infections and ultimately negate the lifesaving benefits of these drugs.

It should also be noted that the correlation between corticosteroid use and incidence of fungal infections in hospitalized COVID-19 patients may be masked by other co-founding risk factors for fungal infections, such as the patient’s history of pulmonary disease, comorbidities and mechanical ventilation [7,109]. For example, in the multicenter study by White et al. [7], apart from corticosteroid use, a history of chronic respiratory disease significantly increased the likelihood of aspergillosis. Importantly, with the exception of the study by Ho et al. [104], most of the currently reported investigations on the potential role between immunosuppressants and fungal infections are from a small cohort of patients. Such a small study size lacks sufficient statistical power and may consequently lead to false conclusions. Indeed, thorough metanalyses of additional retrospective and randomized control studies will help elucidate the role of immunosuppressants in predisposing COVID-19 patients to fungal co-infections.

## 5. Conclusions

Fungal co-infections are reported in severely ill COVID-19 patients admitted to the ICU, with a higher rate of incidence for aspergillosis followed by candidemia, as observed from our literature analysis. Risk factors for such fungal co-infections in the ICU are related to host factors, medical procedures and therapeutics such as corticosteroids, which are designed to alleviate the COVID-19 disease condition. Thus, the links between fungal infections and higher mortality rates in COVID-19 patients and between treatment options and fungal infections provide a conundrum that challenges current medical practices to develop innovative strategies for limiting secondary nosocomial fungal infections in the hospital setting. However, in the interim, prompt diagnosis of fungal co-infections and antifungal administration may help improve prognosis in hospitalized COVID-19 patients.

## Figures and Tables

**Figure 1 jof-07-00545-f001:**
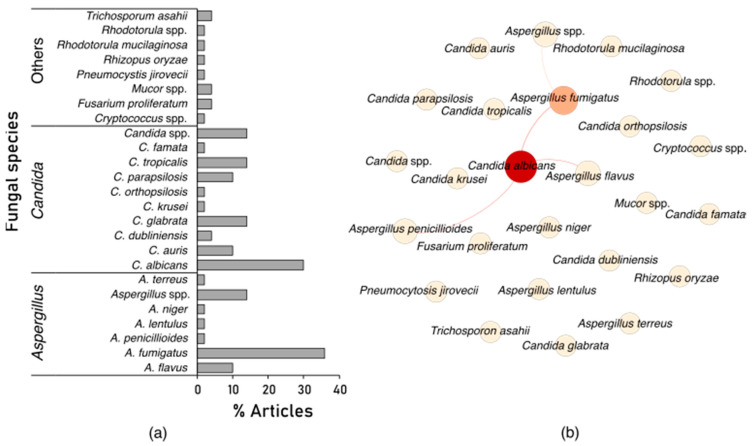
Distribution (**a**) and co-occurrence (**b**) of fungal species across randomly selected 50 case studies, prospective and retrospective published reports on fungal co-infections in hospitalized COVID-19 patients [5,7,9,13,14,15,16,17,18,19,20,21,22,23,24,25,26,27,28,29,30,31,32,33,34,35,36,37,38,39,40,41,42,43,44,45,46,47,48,49,50,51,52,53,54,55,56] (also see Appendix A). Searches were done on PubMed (Medline) and Google Scholar using query terms such as “Co-infections”, “coinfections”, “fungi” or “Fungal infections” in combination with either “COVID-19” or “SARS-CoV-2”. Most (>50%) of the selected articles included cases from the onset of the COVID-19 pandemic in several countries (December 2019–April 2020). In the co-occurrence network, nodes are colored based on betweenness centrality (species with high co-occurrence), while edges (connecting lines) indicate significant (*p* < 0.05) co-occurrence between species. The network was constructed using the visNetwork package of R (https://cran.r-project.org/web/packages/visNetwork/index.html, accessed on 21 May 2021) and further annotated in Gephi (https://gephi.org/, accessed on 21 May 2021).

**Table 1 jof-07-00545-t001:** Summary of literature on potential links between immunosuppressants and fungal co-infections in severely ill COVID-19 patients.

Study Type	City, Country	Cohort Size	Immunosuppressants Therapy for COVID-19	Associated Fungal Infection/Species	Observation with Regards to Corticosteroid Use	Comorbidities and Other Risk Factors	Reference
Case series	Salvador, Brazil	2	Unspecified	*Candida auris*	Prolonged corticosteroid therapy (34 days)	Deep-seated venous thrombosis (Patient 1); Chronic renal insufficiency and dialysis, diabetes mellitus, and hypertension (Patient 2)	de Almeida [30]
Case series	Nuevo Leon, Mexico	12	Hydrocortisone, methylprednisolone, dexamethasone	*C*. *auris*	Corticosteroid treatments preceded the onset of *C*. *auris* infection. Association not particularly investigated	Obesity, asthma, high blood pressure, diabetes, coronary artery disease; valvular heart disease	Villanueva-Lozano et al. [25]
Case series (retrospective)	Porto Alegre, Brazil	11	Prednisone, hydrocortisone, methylprednisolone, dexamethasone	Candidemia (*Candida* spp.)	All cases of candidemia (a 10-fold increase in frequency) in COVID-19 patients occurred after the use of high-doses of corticosteroids	Diabetes, HIV-positive, central venous catheters. Additional risk factors for candidemia were virtually absent	Riche et al. [82]
Prospective study	Madrid, Spain	8	Not specified	Aspergillosis (*A*. *fumigatus*)	Aspergillosis affected mostly (75%) non-immunocompromised COVID-19 patients receiving corticosteroids	Obesity, HTA, COPD, CKD, diabetes mellitus; Mostly (75%) non-immunocompromised patients	Machado et al. [105]
Prospective cohort	Milan, Italy	21	Prednisone, immunomodulators (tocilizumab, mavrilimumab, anakinra, reparixin, and sarilumab) and immunosuppressants (tacrolimus, cyclophosphamide and mycophenolate)	Candidemia (*Candida albicans*, *Candida* spp.)	A higher proportion of candidemia present in COVID-19 patients in the ICU and on immunosuppressive agents	Diabetes, broad-spectrum antibiotics, HIV, etc.	Mastrangelo et al. [57]
Prospective	Milan, Italy	43	Tocilizumab	Candidemia (*C*. *albicans*, *C*. *tropicalis*, *C*. *parapsilosis*)	6.9% prevalence of candidemia observed	Previous hospitalisation in ICU; central venous catheter	Antinori et al. [34]
Case series	Puducherry, India	10	Dexamethasone	Orbital mucormycosis (*Mucor* spp. and *Rhizopus* spp.)	Five patients developed diabetic ketoacidosis after the initiation of corticosteroid therapy for COVID-19 disease	Diabetes mellitus	Sarkar et al. [72]
Case study	Udine, Italy	1	Dexamethasone, tocilizumab	Pulmonary aspergillosis (*A*. *fumigatus*)	*A*. *fumigatus* was isolated 22 days after tocilizumab administration	HBV-related liver cirrhosis, arterial hypertension and mild obesity	Deana et al. [106]
Case study	Brazil	23	Methylprednisolone, prednisone	Candidemia (*C*. *parapsilosis*, *C*. *tropicalis*) and *Trichosporon asahii* fungemia	Fungemia was observed in all patients with a history of prolonged corticosteroid therapy	CVC, exposure to broad-spectrum antibiotics, prior echinocandin therapy, obesity, diabetes	de Almeida Jr [46]
Case series	Paris, France	145	Tocilizumab, sarilumab, hydrocortisone succinate	IPMI (*A*. *fumigatus*; *Fusarium proliferatum*)	Corticosteroid therapies were related to an increased risk for developing IPMI (odds ratio, 8.55; IQR, 6.8–10.3; *p* = 0.01)	HTA, overweight/obesity, diabetes mellitus, COPD. Solid organ transplantation was related to an increased risk for IPMI	Fekkar et al. [107]
Prospective	Wales, UK	135	Prednisolone, methylprednisolone, hydrocortisone, dexamethasone, fludrocortisone	Aspergillosis, yeast infections (mainly *Candida*, one case of *Rhodotorula* fungaemia)	High-dose corticosteroid use increased the likelihood of aspergillosis	Previous chronic respiratory disease also linked to aspergillosis. Associations between comorbidities /underlying conditions and yeast infections were not significant	White et al. [7]
Retrospective	New York, USA	4313	Methylprednisolone, prednisone, dexamethasone, hydrocortisone	Not reported	Corticosteroid use was not associated with increased bacteraemia or fungaemia compared to non-corticosteroid users when administered within the first 7 days	Hypertension, diabetes, CKD, asthma, COPD	Ho et al. [104]
Retrospective cohort study	Paris, France	21	Dexamethasone	Aspergillosis (*Aspergillus* spp.)	Although not statistically significant, a trend was observed between high-dose (≥100 mg) dexamethasone and incidence of IPA	Medical history did not significantly affect IPA	Dellière et al. [88]
Retrospective	India	6	Prednisolone, dexamethasone, or methylprednisolone	Rhino-orbital mucormycosis (*Mucor* spp.)	Five patients developed mucormycosis after treatment with corticosteroids. Mean duration between diagnosis of COVID-19 and development of symptoms of mucor was 15.6 ± 9.6 (3–42) days	Type 2 diabetes	Sen et al. [68]
Retrospective	Chicago, USA	111	Tocilizumab	Fungal pneumonia and sinusitis	Administration of tocilizumab was associated with a higher risk of fungal (*p* = 0.112) infections	Diabetes mellitus, HTA, obesity COPD, cardiovascular disease	Kimmig et al. [103]
Case control study	Barcelona, Spain	71 cases, 142 controls	Tocilizumab, baricitinib, anakinra, dexamethasone, prednisone, hydroxycortisone	*Candida* spp., *Aspergillus* spp., *Fusarium* spp.	Immunomodifiers did not influence occurrence of nosocomial infections in COVID-19 patients	Chronic liver disease, obesity, smoking, invasive mechanical ventilation, hydroxychloroquine	Meira et al. [108]

CKD, Chronic kidney disease; COPD, Chronic obstructive pulmonary disease; CVC, central venous catheter; HTA, Hypertension; HBV, Hepatitis B virus; HBD, High blood pressure, ICU, Intensive care unit; IPA, Invasive pulmonary aspergillosis; IPMI, Invasive pulmonary mold infections.

## Data Availability

Supporting data for Figure 1 is in the Appendix A.

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
