# Peer review of "Risk Factors for Fungal Co-Infections in Critically Ill COVID-19 Patients, with a Focus on Immunosuppressants"

_jof, 2021, doi:10.3390/jof7070545_

Round 1
Reviewer 1 Report
The manuscript of Ezeokoli et al, describes the main risk factors associated with fungal infections in critically ill patients with SARS-CoV-2 infection, with an adequate review of the literature. It focuses mainly on immunomodulation factors caused by treatments (steroids, anti-IL 6...). Perhaps the conclusions should be more concise on the aspect described in the review, but after a few minor changes, it would be a suitable candidate for publication.
Point 1. In the Introduction, in Line 28 you describe the risk of opportunistic infection in COVID-19. After that, in line 31 you cite some reports of fungal co-infections. Maybe it would be better to explain it in only one paragraph for clarification.
Point 2: Line 29: Define SARS-CoV-2 here, as you define in line 140.
Point 3: Line 38: Please define CAPA
Point 4: In line 39 you specify Candida albicans in the same paragraph as Aspergillus fumigatus invasive pulmonary infection, so may be confuse. Specified the type of infection (pulmonary aspergillosis / invasive candidiasis or candidemia) or the pathogen (Aspergillus fumigatus or Candida albicans).
Point 5: In section 3 regarding the risk factors for fungal infections in critically ill COVID-19 patients, it would be useful to clearly differentiate between the two main entities: candidemia/ invasive candidiasis and aspergillosis-CAPA).
Point 6: Line 115: Better use pathogens than microbes
Point 7: Line 126 - 131: This paragraph is a little bit confuse. I understand that what you want to explain is that resistance to antifungals used as first-line prophylaxis can make the infection progress further because we are not properly treating it and breakthrough infections can develop. In the specific case of aspergillosis in COVID, this has not been the case as prophylaxis guidelines have not yet been established in patients who are not immunocompromised. In addition, after that, you talk about antimicrobial resistance and active monitoring for antimicrobial resistance, more used in bacteria than in fungal infections. Please clarify this section and try not to combine the concepts of antifungal resistance with antimicrobial (also in Conclusions sections.
Point 8: In Table 1, column “Associated fungal infection / species”, information about the entity or species is missing in some reports. Where such information is provided, please update the data.
Author Response
REVIEWER 1
Comment: The manuscript of Ezeokoli et al, describes the main risk factors associated with fungal infections in critically ill patients with SARS-CoV-2 infection, with an adequate review of the literature. It focuses mainly on immunomodulation factors caused by treatments (steroids, anti-IL 6...). Perhaps the conclusions should be more concise on the aspect described in the review, but after a few minor changes, it would be a suitable candidate for publication.
Response: Thank you very much for kind comments and suggestions on how the manuscript can be improved. Amongst other major revisions, we have also revised the conclusion to make it more focused on the topic.
Comment- Point 1. In the Introduction, in Line 28 you describe the risk of opportunistic infection in COVID-19. After that, in line 31 you cite some reports of fungal co-infections. Maybe it would be better to explain it in only one paragraph for clarification.
Response: Thanks for pointing this out. Indeed, a paragraph should ideally contain one general theme/idea. We have merged the paragraphs together and provided further text to clarify the subject and emphasis the incidence of mycoses in COVID-19 patients (Line 29-43 of the revised manuscript)
Comment- Point 2: Line 29: Define SARS-CoV-2 here, as you define in line 140.
Response: Thank you for pointing this out. We have now defined SARS-CoV-2 in line 29 of the revised manuscript and have retained only the abbreviations in the subsequent instances (e.g., lines # 39 and 160 of the revised manuscript).
Comment- Point 3: Line 38: Please define CAPA
Response: We have now defined CAPA as “COVID-associated pulmonary aspergillosis” in line 39 of the revised manuscript. Thank you.
Comment- Point 4: In line 39 you specify Candida albicans in the same paragraph as Aspergillus fumigatus invasive pulmonary infection, so may be confuse. Specified the type of infection (pulmonary aspergillosis / invasive candidiasis or candidemia) or the pathogen (Aspergillus fumigatus or Candida albicans).
Response: Thank you pointing out this important distinction and/or clarification. We have reworked the portion of the manuscript to clearly assign the disease with the corresponding pathogen (Line #49-51 of the revised manuscript).
Comment—Point 5: In section 3 regarding the risk factors for fungal infections in critically ill COVID-19 patients, it would be useful to clearly differentiate between the two main entities: candidemia/ invasive candidiasis and aspergillosis-CAPA).
Response: Thank you for this important suggestion on how to make our message clear. We have created two subsections each for CAPA and COVID-19-associated candidiasis (Section 3, 3.1 and 3.2; lines 132-199).
Comment—Point 6: Line 115: Better use pathogens than microbes
Response: We have replaced “microbes” with “pathogens”, although this section has now been deleted/reworked.
Comment—Point 7: Line 126 - 131: This paragraph is a little bit confuse. I understand that what you want to explain is that resistance to antifungals used as first-line prophylaxis can make the infection progress further because we are not properly treating it and breakthrough infections can develop. In the specific case of aspergillosis in COVID, this has not been the case as prophylaxis guidelines have not yet been established in patients who are not immunocompromised. In addition, after that, you talk about antimicrobial resistance and active monitoring for antimicrobial resistance, more used in bacteria than in fungal infections. Please clarify this section and try not to combine the concepts of antifungal resistance with antimicrobial (also in Conclusions sections).
Response: We have reworked the paragraph and have refrained from referring to antimicrobial resistance in the manner akin to bacterial infections. We have also edited the conclusion to keep it concise and in line with the scope of the review. Thank you very much.
Comment—Point 8: In Table 1, column “Associated fungal infection / species”, information about the entity or species is missing in some reports. Where such information is provided, please update the data.
Response: We have reviewed the publications and have updated the information where “species” is reported. Specifically, information in the “) column has been updated for the studies by Mastrangelo et al., Antinori et al., Sarkar et al., Delliere et al., Sen et al. and Kimmig et al. Thank you very much for helping to improve the quality of our manuscript.
Reviewer 2 Report
I would like to congratulate the authors for such a nice work. I find quite relevant at this stage to have a review summarising briefly all the mycoses associated to COVID-19 and here we can see it. However, there certain aspects that need to be considered, in my opinion, by the authors:
MINOR COMMENTS
Figure 1A
- I would suggest to include “spp.” after “Mucor”, “Rhodotorula” and “Cryptococcus”
- There is a misspelling, please correct to “Trichosporon asahii”
- May I ask why “Candida orthopsilosis”, “C. krusei” and “C. dubliniensis” are not included here but they are in figure 1B?
Figure 1B
- I guess that instead of “Penicillium jiroveci” authors meant “Pneumocystis jirovecii”
- There is a misspelling, please correct to “Trichosporon asahii”
- I would suggest to include “spp.” after “Mucor” and “Cryptococcus”
Main text:
- Line 64: ICU abbreviation to line 58
- Line 86: Mucormycosis stands for an infection of moulds from the order Mucorales, not only the genus Mucor.
- Line 99: “Aspergillus” in italics
- May I suggest to include the publications used for your publication in the references section and not only in the supplementary table?
Supplementary table:
- I would suggest to skip the variable continent. In case you want to keep it, I suggest to include it somehow combined: one variable country, one variable continent.
- Wales, so far, is not a sovereign state, but a constituent country of the United Kingdom
- Since the name of the location might not be available in every publication, I suggest to provide only the name of the country
- Study 26: Cologne is a city from Germany, not Italy
- Regarding the name of the pathogen, I suggest to combine all the variables in only 1: Pathogen, in case of a coinfection, you can state such case.
- If these are the publication used for your results, may I suggest that you include them in your reference section?
MAJOR COMMENTS
I really liked the idea of a review summarising all the fungal infections associated to COVID-19. However, there are some aspects from this I consider they need to be reviewed.
First, I find the review a bit unstructured, different aspects of different pathogens are described within the same paragraph, for instance, aspergillosis with candidiasis/candemia. I suggest to structure more precisely the text, that will help a lot to better understand the whole idea and it will allow to have somehow sections depending on the infection.
Second, I miss at least a paragraph, if not a whole section, regarding the methodology you followed for the review: bibliographic research, publication filtering, time of research… I think this is a really important part in every publication, specially in reviews like this.
Third, I find there are many relevant publications on CAPA (aspergillosis) or CAM (mucormycosis) missing. Even some pathogens that have caused infections after COVID-19 are missing: Fusarium spp.
Forth, in some publications, both included in this manuscript and not included, there are incidence rates given, I suggest to use them (Bartoletti et al, or Salmanton-García et al. for CAPA) in order to provide an overview on the magnitude of the different mycoses.
Fifth, in CAPA at least, different definitions have been proposed, however, none has been mentioned in this manuscript.
Author Response
REVIEWER 2
Comment: I would like to congratulate the authors for such a nice work. I find quite relevant at this stage to have a review summarising briefly all the mycoses associated to COVID-19 and here we can see it. However, there certain aspects that need to be considered, in my opinion, by the authors:
Response: We are happy to know that you consider our work to be nice. We also appreciate the constructive criticism provided and have tried to address the comments as much as possible. We tried to summarise the literature on fungal co-infections reported in COVID-19, especially over the early days of the disease (Onset to September, 2020).
MINOR COMMENTS
Comments on Figure 1A:
Comment: I would suggest to include “spp.” after “Mucor”, “Rhodotorula” and “Cryptococcus”.
Response: “Spp.” has been included in the revised manuscript. Thank you.
Comment: There is a misspelling, please correct to “Trichosporon asahii”
Response: Thanks for pointing this out. The spelling has been corrected.
Comment: May I ask why “Candida orthopsilosis”, “C. krusei” and “C. dubliniensis” are not included here but they are in figure 1B?
Response: We omitted them due to their low occurrence in literature compared with other Candida spp. We have now included them for agreement between Figure 1a and 1b.
Comments on Figure 1B
Comment: I guess that instead of “Penicillium jiroveci” authors meant “Pneumocystis jirovecii”
Response: We have revised the species name and have corrected it. Thank you very much. We have revised all the fungal names to ensure no mistakes.
Comment: There is a misspelling, please correct to “Trichosporon asahii”
Response: The spelling has been corrected to “Trichosporon asahii” here and throughout. Thank you.
Comment: I would suggest to include “spp.” after “Mucor” and “Cryptococcus”
Response: We have included “spp.” after “Mucor” and “Cryptococcus” as suggested.
Comments on the Main text:
Comment—Line 64: ICU abbreviation to line 58
Response: The abbreviation “ICU” has been moved to line 58. The plural variant “ICUs” has been earlier defined in line 30.
Comment—Line 86: Mucormycosis stands for an infection of moulds from the order Mucorales, not only the genus Mucor.
Response: This has been corrected (Line 137). Thank you.
Comment—Line 99: “Aspergillus” in italics
Response: We have ensured that all Latin names are italicised. In this instance, we substituted “Aspergillus” for “fungal” (Line 160).
Comment—May I suggest to include the publications used for your publication in the references section and not only in the supplementary table?
Response: Thanks for the suggestion. We have now included these references in the reference list of the revised manuscript (Figure 1 footnote; Line 73).
Comments on Supplementary table:
Comment: I would suggest to skip the variable continent. In case you want to keep it, I suggest to include it somehow combined: one variable country, one variable continent.
Response: The variable continent has been removed as it is not very informative.
Comment: Wales, so far, is not a sovereign state, but a constituent country of the United Kingdom
Response: Thank you for calling our attention to this. We have used “UK” instead.
Comment: Since the name of the location might not be available in every publication, I suggest to provide only the name of the country
Response: Thank you. We have provided only the name of the country.
Comment: Study 26: Cologne is a city from Germany, not Italy
Response: Thanks for pointing out this mistake. We have removed the city information and retained only the country information as suggested in the earlier comment. In this instance, “Germany” has been indicated.
Comments: Regarding the name of the pathogen, I suggest to combine all the variables in only 1: Pathogen, in case of a coinfection, you can state such case.
Response: We have combined this information as suggested and mentioned cases of co-infection where applicable. The previous outlay was the raw data from which Figure 1 was prepared. The current version is indeed neat and concise. Thank you.
If these are the publication used for your results, may I suggest that you include them in your reference section?
Response: We have included the references in the reference list of the revised manuscript (Figure 1 footnote; Line 73).
MAJOR COMMENTS
Comment: I really liked the idea of a review summarising all the fungal infections associated to COVID-19. However, there are some aspects from this I consider they need to be reviewed.
Response: We are happy to know that the reviewer likes the idea behind our work. We have provided responses to the specific points raised by the reviewer below.
Comment—First, I find the review a bit unstructured, different aspects of different pathogens are described within the same paragraph, for instance, aspergillosis with candidiasis/candidemia. I suggest to structure more precisely the text, that will help a lot to better understand the whole idea and it will allow to have somehow sections depending on the infection.
Response: We have now structured the paper to highlight the two most commonly reported fungal infections and a section to comment on the “hot topic” of COVID-19-associated mucormycosis.
Comment—Second, I miss at least a paragraph, if not a whole section, regarding the methodology you followed for the review: bibliographic research, publication filtering, time of research… I think this is a really important part in every publication, especially in reviews like this.
Response: Thanks for pointing this out. By no means is our paper a systematic review of the literature as this is an active and evolving area of study (For example, only recently did mucromycosis become commonly reported, and it is considered largely a post-COVID-19 development). In the legend of Figure 1 of the revised manuscript, we have provided a definition or boundary for the search term and also provided the references used for Table S1 in the main body of the paper. We tried to include as many papers as possible but because this is an evolving situation some papers might have been missed. As highlighted in the title of the paper, our focus was on the role of immunosuppressants in fostering fungal co-infections. However, to introduce the subject and demonstrate a measure of in-depth study of the literature, using Figure 1, we summarised our findings from 50 randomly selected papers reporting fungal co-infections. We have also stated that the 50 reports included in our paper are non-exhaustive.
Comment—Third, I find there are many relevant publications on CAPA (aspergillosis) or CAM (mucormycosis) missing. Even some pathogens that have caused infections after COVID-19 are missing: Fusarium spp.
Response: Thank you for pointing this out. As mentioned in the earlier comments, indeed, our work is not an exhaustive systematic review as earlier mentioned. We have revised the manuscript and included additional 10 articles (we removed 5 articles from the previous list that did not specify at least the genus of the fungi identified in the patients. We have also talked about CAM in section 2 that deals with “Fungal co-infections in COVID-19 disease”.
Comment—Forth, in some publications, both included in this manuscript and not included, there are incidence rates given, I suggest to use them (Bartoletti et al, or Salmanton-García et al. for CAPA) in order to provide an overview on the magnitude of the different mycoses.
Response: We have included incidence rates in the introduction for an appreciation of the magnitude of the different mycoses (Lines 31-39).
Comment—Fifth, in CAPA at least, different definitions have been proposed, however, none has been mentioned in this manuscript.
Response: The definitions of CAPA is not within the scope of this review, but we understand why it might be worth mentioning. In the revised manuscript, we have briefly pointed the readers to relevant literature dealing with the definitions for CAPA and have provided basic definitions of a proven CAPA case and candidiasis/candidemia. These have been addressed in Section 3 (lines 171-183 and 209-210).
Round 2
Reviewer 2 Report
All the open queries have been solved, I suggest to accept the manuscript.